# Initiating and imaging the coherent surface dynamics of charge carriers in real space

K.R. Rusimova[1,2], N. Bannister[1], P. Harrison[1], D. Lock[1], S. Crampin[1], R.E. Palmer[2] & P.A. Sloan[1]

The tip of a scanning tunnelling microscope is an atomic-scale source of electrons and holes. As the injected charge spreads out, it can induce adsorbed molecules to react. By comparing large-scale 'before' and 'after' images of an adsorbate covered surface, the spatial extent of the nonlocal manipulation is revealed. Here, we measure the nonlocal manipulation of toluene molecules on the Si(111)-7 $\times$ 7 surface at room temperature. Both the range and probability of nonlocal manipulation have a voltage dependence. A region within 5–15 nm of the injection site shows a marked reduction in manipulation. We propose that this region marks the extent of the initial coherent (that is, ballistic) time-dependent evolution of the injected charge carrier. Using scanning tunnelling spectroscopy, we develop a model of this time-dependent expansion of the initially localized hole wavepacket within a particular surface state and deduce a quantum coherence (ballistic) lifetime of $\sim$10 fs.

[1] Centre for Nanoscience and Nanotechnology, Department of Physics, University of Bath, Bath BA2 7AY, UK. [2] Nanoscale Physics Research Laboratory, School of Physics and Astronomy, University of Birmingham, Birmingham B15 2TT, UK. Correspondence and requests for materials should be addressed to P.A.S. (email: p.sloan@bath.ac.uk).

During imaging, the scanning tunnelling microscope (STM) injects charge into a surface on average every 1 nanosecond. Less than 1 picosecond after injection a charge carrier will have undergone a hierarchy of scattering events leading to eventual thermalization with the surface. In this interval, a charge carrier may transfer energy to individual atoms and molecules on the surface and induce, for example, bond breaking[1] or atomic desorption[2]. Thus, atomic manipulation is the signature of the underlying dynamics of the charge carriers. Conventional atomic manipulation is restricted to the tunnel junction and the time-scale of the charge dynamics has to be inferred from the complementary calculations[3]. Conversely, nonlocal manipulation, in which molecules located at some tens of nanometres[4–8] away from the tunnelling site respond to charge injection, can expose the $\sim 200$ femto-second (fs) diffusive dynamics of injected electrons[9].

Here, we employ the nonlocal manipulation technique at room temperature to probe the ultrafast ballistic dynamics of the injected charge carrier, that is, while it coherently evolves from its original quantum state. We find a coherence length scale of up to 15 nm, and deduce an associated coherence lifetime of $\sim 10$ fs. Recent advances in the ability to construct atomic-scale structures by conventional atomic manipulation have opened the way to atomic architectures that exhibit engineered quantum effects on metals[10,11], semiconductors[12,13] and even graphene[14]. Our results indicate the possibility of harnessing such quantum behaviour well above normal 4 K operating temperatures, possibly reaching the more practical room temperature regime.

**Figure 1 | Nonlocal manipulation of toluene molecules on Si(111)-7 × 7.** Nonlocal manipulation of toluene molecules bonded to unfaulted-middle adatom sites on the Si(111)-7 × 7 surface at room temperature. STM images (30 × 30 nm, +1 V, 100 pA) of toluene on Si(111)-7 × 7 taken before (**a–c**) and after (**d–f**) an injection of charge at the unfaulted-middle adatom site 'X': (**a**) injection at −1.6 V, 900 pA and 45 s; (**b**) −1.9 V, 900 pA and 10 s; (**c**) −2.2 V, 900 pA and 10 s. (**g–i**) Corresponding radial distributions of the fraction of manipulated molecules. Fits to the data are: blue-dashed line, 2D diffusion with single decay channel (only fitted to purple circle data points); solid-red line, two-step inflation and diffusion model fitted to all data points. Error bars are the standard error of five injection experiments at each injection bias voltage.(**j**) High-resolution STM image with Si(111)-7 × 7 unit cell outlined and a single molecule bonded to an unfaulted-middle adatoms site (black-spot). (**k**) Schematic showing the crystal structure of (**j**) with the bonding site of the toluene indicated and adatoms labelled: UM, UC, FM and FC.

## Results

**Nonlocal manipulation of toluene molecules with holes.** Figure 1 shows three pairs of $30 \times 30$ nm STM images taken before and after the injection of holes (negative sample bias) at the location marked 'X' at voltages (A,D) $-1.6$ V, (B,E) $-1.9$ V and (C,F) $-2.2$ V. The bright spots are silicon adatoms of the Si(111)-7 $\times$ 7 surface and the dark-spots indicate chemisorbed toluene molecules[15]. Before hole injection, there is a random distribution of toluene molecules (see Supplementary Fig. 1 for full-scale $50 \times 50$ nm version of Fig. 1a,d), while after injection, there is a markedly lower toluene (dark-spot) coverage surrounding the injection site. We quantify this nonlocal desorption (manipulation) behaviour by identifying the crystallographic location[16] relative to the injection site of each molecule before and after the injection. Figure 1g–i show, for each voltage, the radial dependence of the ratio of molecules that are manipulated (that is, leave their initial adsorption site), $N(r)$, to the total number of molecules that were originally at that radial distance, $N_0(r)$. Here, we inject into unfaulted-middle (UM) sites of the Si(111)-7 $\times$ 7 surface (Fig. 1j,k) and report the behaviour of molecules initially bonded to crystallographically equivalent UM sites (other sites produce similar behaviour—see Supplementary Figs 3–5 and 7–9).

**'Suppression' of manipulation close to injection site.** At radial distances $>15$ nm, the radial distributions are well described by a one-hole manipulation process coupled to a 2 dimensional (2D) diffusive charge-transport model with a single decay channel, see Supplementary Fig. 2 (ref. 9). (The manipulation of the molecule, that is, induced movement away from the original binding site, could in principle be due either to a phonon emission during this decay or a non-adiabatic decay channel[17]). The diffusive model parameters are $\beta$, the probability per generated hole of inducing a manipulation event, and $\lambda$, the range of the diffusive transport. The blue-dashed line in Fig. 1g–i show fits of this model to the data, and Fig. 2a,b show the bias dependence for these parameters. The principal result of this work is that at distances less than 15 nm there is a strongly voltage-dependent 'suppression' region $R$, where the probability of manipulation is far below the predicted value. Furthermore, the probability $\beta$ of inducing a manipulation event increases monotonically with increasing negative voltage (Fig. 2a), whereas the range $R$ of the suppression region has a saw-tooth behaviour and returns to a minimum value at a voltage of $-1.5$ V. The same voltage marks a transition between near constant values for the diffusion length scale. (Similar behaviour is found for the other molecular binding sites of the Si(111)-7 $\times$ 7 surface unit cell, Supplementary Figs 3–5 and there is evidence of a near identical voltage discontinuity for STM-induced nonlocal desorption of chlorine atoms from the Si(111)-7 $\times$ 7 surface by holes[18]).

**Connection between surface states and transport regions.** We associate the plateau region of the diffusion length $\lambda$ with diffusive charge transport through a particular prominent surface state[9,19], and identify the threshold voltage corresponding to each plateau region with the onset of particular surface states. This assignment is supported by scanning tunnelling spectroscopy (STS) measurements of UM silicon adatoms (the injection site) shown in Fig. 2d. The nonlocal threshold at $-1.2$ V corresponds to the onset of the $S_3$ back-bond surface state[20], which peaks at $-1.6$ V in Fig. 2d, and the second threshold at $-1.6$ V corresponds to the onset of a state that peaks at $-2.2$ V[18], which for consistency we label $S_4$. Thus, the nonlocal manipulation thresholds are matched with the onsets of particular surface states, not their peak-positions. We therefore

label the two charge-transport regimes as $T_{S_3}^{UM}$ between $-1.2$ and $-1.5$ V and $T_{S_4}^{UM}$ for manipulation between $-1.5$ and $-2.3$ V. We find analogous behaviour for the other adsorption sites (Supplementary Figs 3–5).

## Discussion

The discontinuity in the range of the suppression region $R$ with increasing voltage immediately rules out an electric-field effect as the cause of the suppression, since the $E$-field strength monotonically increases with injection voltage (Supplementary Fig. 6B). It also eliminates $E$-field induced dangling-bond charging as a possible 'suppression' mechanism[21]. Furthermore, to maintain a constant current as the voltage increases the, STM tip withdraws from the surface (Supplementary Fig. 6A), and so any mechanical, short-range tip-molecule interaction would be reduced (not enhanced) as the voltage increased.

Instead, here we propose that the relative suppression of the molecular manipulation probability close to the STM tip reflects a (counter-intuitive) reduction of the number of charge carriers that interact, either directly through a short-lived ionic state or

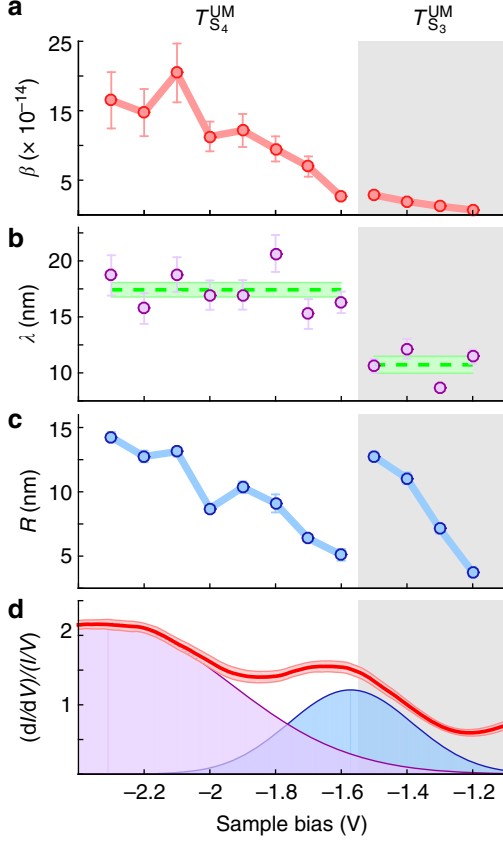

**Figure 2 | Injection bias dependence of nonlocal manipulation of toluene molecules with holes.** Injection bias dependence for injection into UM adatom sites and nonlocal manipulation of UM toluene molecules. (**a**) The probability $\beta$ of manipulation per injected hole, (**b**) diffusion length-scale $\lambda$ with the average length-scale for each transport region indicated with green bar with uncertainty given by the width of the bar. (**c**) The range of the suppression region $R$. (**d**) Red line shows variable gap STS of clean UM adatoms. The standard error (just visible) of 38 individual spectra has been shaded. Two Gaussian functions have been fitted as indicated (the resulting superposition of these states (not shown) follows the measured STS almost exactly): peak position and full width at half maximum (FWHM) $(-1.6 \pm 0.4)$ eV and $(-2.3 \pm 0.9)$ eV. The voltage domains of the transport regimes $T_{S_3}^{UM}$ and $T_{S_4}^{UM}$ are indicated. All errors quoted are one s.d.

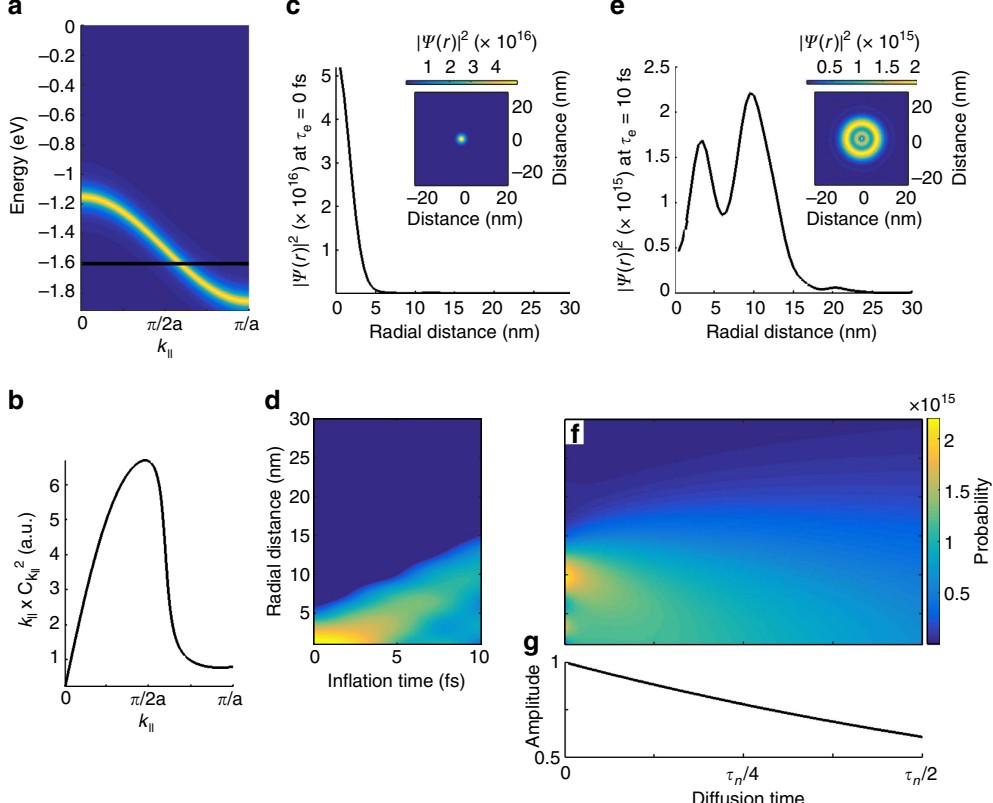

**Figure 3 | Two-step inflation-diffusion model.** Model of two-step, coherent expansion and diffusive transport of injected charge, for a injection bias of $-1.6$ V. (**a**) Energy band diagram of $S_3$ back-bond state. The black line indicates the injection voltage. Intensity is proportional to the density of states (DOS) as a function of energy and momentum. (**b**) Tunnelling probability of populating $k_{||}$ values. (**c**) Radial probability distribution (RPD) of the wavepacket (probability per unit area) at $t = 0$. Inset shows a 2D real-space image of the RPD. (**d**) Time evolution of the RPD up to a time of $t = 10$ fs (for ease of viewing the colour map is on a log scale). (**e**) Linear plot and inset 2D plot of the RPD after 10 fs. (**f**) Diffusive time evolution of the RPD. (**g**) Time evolution of the total charge demonstrating exponential decay.

indirectly as the carrier decays via phonon emission, with the adsorbed molecule at distances close to the injection site: after injection, the probability distribution undergoes rapid ballistic expansion, or 'inflation', followed by relatively slow 2D diffusion. The diffusion is therefore initiated at a radius that is distant from the injection site. The solid radial curves in Fig. 1g–i correspond to such a two-step (ballistic-inflation followed by diffusion) model, using the initial conditions for diffusion an annulus of width 2 nm centred on the injection site at a radius $R$, the range of the suppression region. The excellent fit to the measured data across all length-scales is evident.

Charge injected from the tip of an STM undergoes a series of processes eventually leading to thermalization with the bulk crystal. These follow a hierarchy of time scales: elastic momentum scattering (that is, directional scattering); inelastic scattering within an electronic state to form a quasi-equilibrated distribution (it is during this phase that the nonlocal diffusive transport occurs); inelastic scattering out of the state towards lower lying states where the charge has insufficient energy to induce manipulation (this is the decay channel of the diffusive model). Since the two-step inflation-diffusion model accounts so well for the experimental observation, we conclude that the inflation region is the result of the charge dynamics before quasi-equilibration. Furthermore, since any incoherent scattering, whether elastic or inelastic, would lead to diffusive behaviour, the inflation region must correspond to the dynamics of the injected charge before it undergoes any scattering, while it remains a coherently expanding quantum wavepacket. That is,

the inflation region is determined by the positions that the injected charges undergo their first scattering event.

We develop a simple model to connect the range of the inflation region with the band-structure of the surface and hence access the time-scale of the coherent-inflation process. In line with their prominence in our STS measurement, the inflation process is modelled on the initial occupation (by tunnelling) of an electronic surface state. The state is modelled as an azimuthally isotropic 2D state with a tight-binding dispersion $E \propto \cos(k_{||}a)$. Figure 3 shows the model for the $S_3$ state, with the dispersion shown in Fig. 3a. At a particular injection voltage, we compute the probability of tunnelling with a particular wavevector in the surface plane $k_{||}$ (Fig. 3b) and use these values to construct a time-dependent cylindrically symmetric wavepacket. Figure 3c shows the radial probability distribution of the initial wavepacket. Figure 3d shows the time evolution of the wavepacket until the end of the inflation time at 10 fs, and Fig. 3e shows the radial extent of the wavepacket at the end of the inflation period. The spread of the wavepacket from the injection site is evident. This radial distribution is used as the starting point for diffusion (Fig. 3f) with an exponential decay (Fig. 3g).

The time-integrated radial distribution provides the total number of charges that can interact with a molecule at a particular distance. Figure 4 shows a series of these curves fitted to voltage-dependent experimental data. Within a transport region each curve has (i) the same diffusive parameters derived from Fig. 2 ($T_{S_3}^{UM}$ $\lambda = (9.0 \pm 0.2)$ nm (note all errors are s.e.m.), $T_{S_4}^{UM}$ $\lambda = (18.0 \pm 0.3)$ nm); (ii) the same fitting parameter $\tau_i$ for

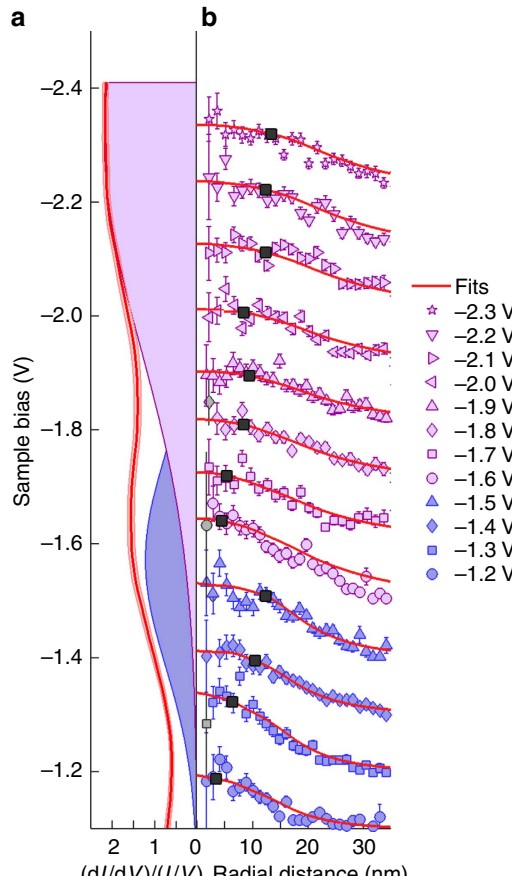

**Figure 4 | Comparison of STS with nonlocal manipulation and the inflation-diffusion model.** Comparison of STS with nonlocal manipulation and the inflation-diffusion model. (**a**) Red curves give the STS spectrum taken on a clean UM site. Shaded areas show Gaussian fits to the two peaks: peak position and FWHM $(-1.6 \pm 0.4)$ eV and $(-2.3 \pm 0.9)$ eV. (**b**) Injection voltage dependence of nonlocal manipulation of UM toluene molecules with a UM injection site. Radial distribution curves have been vertically offset to aid clarity and match STS energy axis. Solid-red lines show the inflation-diffusion model fitted to each dataset (light grey data points were omitted from the fits). Black markers indicate the range of inflation region determined form the experimental data (as given in Fig. 2c). Error bars are the standard error of five injection experiments at each injection bias voltage.

the inflation time, and (iii) a voltage-dependent amplitude. The excellent global fits of the inflation-diffusion model give inflation times for $T_{S_3}^{\mathrm{UM}}$ $\tau_i = (11 \pm 1)$ fs and $T_{S_4}^{\mathrm{UM}}$ $\tau_i = (9 \pm 1)$ fs. We also find good global fits for the other three possible molecular binding sites and good correlations with their STS measurements (Supplementary Figs 7–9). We conclude that the nonlocal STM manipulation measurements capture the initial coherent dynamics of the injected charge carriers.

The inflation time represents the coherent or ballistic transport regime. We find no direct comparison in the literature, but the relaxation time of electrons within the dangling bond $U_1$ state has been measured to be ~40 fs[22]. This is longer than our ~10 fs time-scale and, may be due to the $U_1$ state lying within the bulk band-gap unlike our higher-lying states. However, recent calculations for holes in bulk silicon giving relaxation times of ~10 fs[23] have demonstrated the sensitivity to both the identity of the band that the electronic excitation resides in, and to the energy of the electronic excitation within the band. We also note, again for bulk states, that the mean free paths computed for holes

in Si are in the tens of nanometres range, mirroring our surface-sensitive measurements.

The STM injects charges with a range of energies, from the bias voltage level to the Fermi level. Figure 3b shows this relationship in terms of parallel momenta. Our experiments are therefore a weighted average over a range of energies and hence, according to Bernardi et al.[23], a range of coherent lifetimes. In our simple scheme, we fit a single inflation (coherence) lifetime to each surface state. The site-to-site variation of the lifetimes we obtain may reflect the spatial distribution of the particular surface states within a Si(111)-7 × 7 unit cell giving rise to differing site-to-site coherent propagation. Phonon amplitudes will also vary within the large 7 × 7 unit cell[24], so that $e$-phonon relaxation rates can be expected to be site-specific.

The coherent-inflation time itself most likely has two main components, a temperature-independent coupling to bulk states and a temperature-dependent phonon scattering. Our earlier work with temperature-dependent electron injection found no obvious increase in the inflation region at lower temperatures, suggesting that the measured inflation region is an intrinsic property of the surface state. It therefore offers the possibility (as in the case of $C_{60}$ (ref. 25) and other molecular overlayers[26]) of designing a system with reduced coupling to bulk states and hence an increased quantum coherent-inflation range, thereby extending the range over which a possible quantum device could work.

## Methods

**Sample preparation and imaging.** All experiments were performed at room temperature with an Omicron STM1 in an ultrahigh vacuum chamber at base pressure $1 \times 10^{-10}$ mbar. Tungsten tips were electrochemically etched in a 2 M NaOH solution and out-gassed by resistive heating in high vacuum. Pre-cut silicon samples from an $n$-type phosphorus doped Si(111) wafer (0.001–0.002 Ω cm) were prepared using an automated flashing routine[19]. Toluene was purified by the freeze-pump-thaw technique with liquid nitrogen and checked for purity with a quadrupole mass spectrometer. Typically, the surface was saturated with a toluene dose of 4 Langmuirs, corresponding to a coverage of 1,500–2,000 molecules in a 50 × 50 nm image. All images were obtained in constant current mode with passive scanning conditions ($+1$ V and 100 pA)[16].

**Nonlocal injections.** Experiments were performed with a Nanonis SPM control system programmed to inject into user-selected atomic locations. Stability during the injections was provided by in-house developed drift-tracking software, and a feature-locking technique employed to ensure the correct injection position. The individual locations of all molecules adsorbed to the surface before and after an injection experiment were determined with respect to the injection site using an in-house computer analysis suite[16]. This resulted in a molecule identification accuracy of more than 99%. The results were corrected for thermally induced desorption using the technique outlined in ref. 16. To ensure good statistical results, the total injected charge (number of holes) was varied so that approximately half the molecules at half the distance to the edge of the full image (50 × 50 nm) were manipulated.

The Si(111)-7 × 7 surface presents four distinct adatoms that a toluene molecule can bind to: faulted corner (FC), faulted middle (FM), unfaulted corner (UC) and unfaulted middle (UM) (Fig. 1j,k). Each site has slightly different electronic properties[27] and binding energies for a chemisorbed toluene molecule[28]. To ensure a consistent experimental scenario here, we only inject into clean UM adatom sites (that is, with no adsorbate molecule) and report in the main text the manipulation of toluene molecules initially bonded to UM sites. Other injection and adsorption sites produce qualitatively similar results (Supplementary Figs 3–5 and 6–9), but the trends in the data are clearest for UM injection and adsorption sites. To further ensure the robust nature of the results, we use an automated experimental system allowing many repeats at each set of injection parameters.

**2D diffusion model.** As derived in ref. 9 the mathematical model for 2D diffusion with a single decay channel gives

$$\frac{N(r)}{N_0(r)} = 1 - \exp\left[ -\frac{n_i \cdot \beta}{2\pi D} K_0\left(\frac{r}{\lambda}\right) \right] \qquad (1)$$

where $n_i$ is the number of generated holes, $D$ is the diffusion constant of the charge carriers and the combined term $\beta$ gives the probability (a cross-section per unit time) that an injected hole will induce a manipulation event. Finally $K_0$ is a

modified Bessel function of the second kind with argument that includes the parameter $\lambda$, which gives the range of the diffusive transport.

**Determination of suppression region $R$.** To accurately determine the size of the suppression region, $R$, the 2D diffusion model was fitted to the data from the largest radial distance to a smaller distance $r_0$. By comparing the coefficient of determination and the fitted value of $\lambda$ as a function of $r_0$, we could identify the minimum value of $r_0$ that gave a coefficient of determination value $> 0.9$, which was the smallest value of $r_0$ that had a near constant $\lambda$ fitting parameter.

**Scanning tunnelling spectroscopy.** Variable gap $dI/dV$ spectra of individual Si adatoms were obtained directly with a lock-in amplifier in the region from 0 to $-2.5\,V$ from an initial stabilizing voltage of $+1\,V$ at 100 pA. The tunnelling gap was reduced by 25 pm V$^{-1}$ to amplify the signal at low bias within the band-gap region. A typical spectrum contained 200 data points each acquired over 150 ms with a lock-in modulation of 20 mV at 521 Hz. The STS curves presented here are the average of 38 such measurements.

**Inflation model: 2D surface state.** For each molecular adsorption site we find site-specific nonlocal manipulation and STS properties, reflecting site-specific coupling to the specific charge-transport surface state. We therefore determine site-specific energy onsets and the bandwidths. We model the 2D surface state, which mediates the coherent expansion as a cosine tight-binding like electronic state with dispersion relation $E = E_0 + \Delta E/2[\cos(ka) - 1]$, where $E_0$ is the bandedge, $\Delta E$ is the bandwidth and $a$ is the Si(111)-7 × 7 unit cell length of 2.68 nm. The energy onset $E_0$ of the state is given by the thresholds found for the various transport regions identified in the diffusion length-scale $\lambda$, see for example Fig. 2b, giving for UM injections and UM molecular manipulations for the S$_3$ state an onset of $-1.15 \pm 0.1\,V$ and for the S$_4$ state $-1.55 \pm 0.1\,V$; for UC molecules, $-1.15$ and $-1.55\,eV$; for FM molecules, $-1.15$ and $-1.35\,eV$; and for FC molecules $-1.15$, $-1.65\,eV$ and a third transport region starting at $-2.15\,eV$. The S$_3$ onsets match with that reported by photoemission for the S$_3$ state and the fact that a section of the ARUPS measured results lie within the projected bulk band-gap is strong evidence for a surface state of the Si(111)-7 × 7 unit cell[29,30]. The S$_3$ state has been found to have downward dispersion[29,31] and be coupled to $p_x$ and $p_y$ orbitals confirming the downwards dispersion.

To determine a state-specific bandwidth $\Delta E$ we use the width of the peaks found on our site-specific STS measurements averaged across all the sites giving a mean peak full width at half maximum (FWHM) of 0.49 eV for S$_3$ and 0.76 eV for S$_4$. In ref. 29 $a \sim 0.35\,eV$ dispersion between the $\bar{M}$ point and half way to the $\bar{\Gamma}$ point is reported, giving a full bandwidth of $\sim 0.7\,eV$. Hence, our measured STS FWHM of 0.49 eV corresponds to a bandwidth of 0.7 eV for S$_3$ and a bandwidth of 1.1 eV for S$_4$. We introduce a Lorentz broadening with a half width at half maximum $\sigma = \hbar/\tau_i$ with $\tau_i$ the inflation lifetime. The resulting energy and momentum spectral density $n(E, k_{\parallel})$ is given by

$$n(E, k_{\parallel}) \propto \frac{\sigma^2}{\sigma^2 + \left(E - E_0 - \frac{\Delta E}{2}\left[\cos(k_{\parallel}) - 1\right]\right)^2}, \tag{2}$$

and shown in Fig. 3a for the S$_3$ state.

**Tunnelling probability.** Charge carriers tunnel across the STM junction with a probability that depends on their energy and lateral momentum $k_{\parallel}$. The standard transmission coefficient for tunnelling[32] is

$$T(E, k_{\parallel}) \propto \exp\left(-2z\sqrt{\frac{2m_e}{\hbar^2}(E_b - E) + k_{\parallel}^2}\right), \tag{3}$$

where we take the barrier height $E_b$ to be the mean height of the barrier of the tunnel junction and assume a vacuum level $E_v$ at 4.6 eV for both sides giving $E_b = E_v + E_i/2$, where $E_i$ is the injection energy.

**Wavepacket construction.** The solutions to the 2D cylindrical Schrödinger equation are the Bessel functions. We construct a time-dependent wavepacket from a weighted sum over $k$-space,

$$\Psi = A \int_{k_{\parallel}=0}^{\pi/a} c_{k_{\parallel}} \cdot \sqrt{\frac{k_{\parallel}}{2\pi}} \cdot J_0(k_{\parallel} \cdot r) \cdot e^{-iE_{\parallel}t/\hbar} \cdot dk_{\parallel} \tag{4}$$

where $A$ is an overall constant of normalization, $\sqrt{k_{\parallel}/2\pi}$ is the normalization constant for each $k_{\parallel}$ state $(\int_0^{\infty} 2\pi r J_0(k_{\parallel} \cdot r) J_0(k'_{\parallel} \cdot r) dr = (2\pi/k_{\parallel})\delta(k_{\parallel} - k'_{\parallel}))$ and $c_{k_{\parallel}}$ is the relative amplitude of the particular $k_{\parallel}$ reflecting the probability of tunnelling into it:

$$c_{k_{\parallel}}^2 = \int_{E_i}^{E_F} T(E, k_{\parallel}) \cdot n(E, k_{\parallel}) dE, \tag{5}$$

where $E_F$ is the Fermi level in the sample and $E_i$ the injection energy (STM bias voltage).

We use the Bessel functions in equation (4) to describe the surface band eigen-functions, since detailed knowledge of the actual wave functions is not available. Strictly speaking, the Bessel functions are only applicable to states exhibiting free-electron like dispersion, which is only the case towards the edges of the surface bands here. To assess the likely impact of this we have also considered inflation of the initial wavepacket within the opposing limit, solving on a hexagonal lattice the time-dependent Schrödinger equation $i\hbar\partial|\psi\rangle/\partial t = H|\psi\rangle$ assuming the tight-binding Hamiltonian,

$$H = \epsilon_0 \sum_i |i\rangle\langle i| - \gamma \sum_{\langle i,j \rangle} |i\rangle\langle j| \tag{6}$$

corresponding to one orbital $|i\rangle$ per Si(111)-7 × 7 unit cell, nearest-neighbour hopping, and with $\epsilon_0$, $\gamma$ appropriate to the known dispersion of the surface band. Supplementary Fig. 10 shows the radial evolution predicted in the two cases, and we see both yield comparable expansion properties, with the state density advancing at similar rates, and being relatively uniformly distributed behind the front. This demonstrates the inflation is relatively insensitive to the precise nature of the assumed surface band wave functions.

**Radial diffusion and decay.** Numerical calculation use an evenly spaced grid of radii up to 300 nm in $\Delta r = 1\,nm$ steps and 400 time-steps $\Delta t$ to a time $2\tau_n$. We use a reduced time $t' = t/\tau_n$, $\Delta t' = \Delta t/\tau_n$ and a reduced distance $r'_i = r_i/\lambda$ and $\Delta r' = \Delta r/\lambda$ to write the standard iterative scheme including an exponential decay term to compute the radial probability distribution $P(r'_i, t'_j)$ of the diffusing charge

$$P(r'_i, t'_{j+1}) = \left\{ P(r'_i, t'_j) + \frac{\Delta t'}{\Delta r'^2}\left[\left(1 + \frac{\Delta r'}{2r'_i}\right)P(r'_{i+1}, t'_j)\right.\right.$$
$$\left.\left. - 2P(r'_i, t'_j) + \left(1 - \frac{\Delta r'}{2r'_i}\right)P(r'_{i-1}, t'_j)\right]\right\}e^{-\Delta t'}. \tag{7}$$

The shape of the time-integrated radial probability density, used below to fit to the experimental data, is independent of the choice of the lifetime $\tau_n$. The lifetime does affect the overall computed probability per hole of inducing manipulation $\beta$. In this work we focus on the shape and range of the nonlocal effect not the absolute probability per hole. Nevertheless, to give consistent and realistic values we use $\tau_n = 200\,fs$ in line with the results found for nonlocal electron manipulation[9].

**Fit to experimental data.** Experimentally we measure the number of molecules that leave their original location $N(r)$ due to an injection of $n_i$ holes and the original number of molecules $N_0(r)$ within an annulus at radius $r$. Hence,

$$\frac{N(r)}{N_0(r)} = 1 - \exp\left[-\beta n_i \cdot \int_0^{\infty} P(r, t) dt\right] \tag{8}$$

See Lock et al.[9] for full derivation. Thus, the only fitting parameters in the two-step inflation-diffusion model are (i) the inflation lifetime $\tau_i$ and (ii) $\beta$, the probability per hole of inducing manipulation. We justify the use of a single inflation time for all voltages within a transport region, since the energy relaxation time for hot holes in bulk silicon is near constant for holes with at least 0.1 eV of excess energy above the band edge[23], and for electrons in the U$_1$ band of the Si(111)-7 × 7 surface with 0.1–0.3 eV of excess energy[22].

**Data availability.** All data supporting this study are openly available from the University of Bath data archive at http://doi.org/10.15125/BATH-00230.

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

## Acknowledgements

P.A.S. gratefully acknowledges support from the EPSRC Grant EP/K00137X/1. K.R.R. was supported by a University of Bath studentship and D.L. by an EPSRC DTA studentship. R.E.P. is grateful to EPSRC for the award of a Fellowship.

## Author contributions

All authors have contributed significantly to this work. K.R.R. performed the nonlocal experiments and data analysis. N.B., P.H. and D.L. performed initial nonlocal experiments. N.B., P.H. and K.R.R. performed and analyzed the STS. S.C. provided theoretical support and performed time-dependent tight-binding simulations. P.A.S. and R.E.P. conceived, designed and led the research.

## Additional information

**Competing financial interests:** The authors declare no competing financial interests.

