## [Peer review file · Nature Communications]

Reviewers' Comments:

Reviewer #1 (Remarks to the Author)

The authors describe a novel approach to infer carrier dynamics based on the effect of holes injected via an STM tip into the Si(111) surface. The "coherent" vs. diffusive regime is inferred by measuring the fraction of desorbed toluene molecule vs. distance from the injection point, as a function of bias (and thus, hole energy).

I think these experiments are striking, highly original, and certainly worthy of publication in nature communications. On the other hand, I am not entirely happy with the explanation provided by the authors. My comments on this point are as follows:

1) The regime called "inflation" in the manuscript is nothing else than the ballistic transport regime in which carriers travel without scattering. The standard terminology is ballistic transport, and the manuscript should be changed accordingly. There is a whole field of ballistic electron microscopy and computation of ballistic mean free paths.

2) I completely agree that the 10-15 nm distances found in Figure 2c for the suppression region are the distance over which carriers travel without scattering. One can compare these distances with the computed energy-dependent mean free paths in silicon, in Figure 3 of reference 23 (Bernardi et al; note the shifted scale for the hole energy in the x-axis in that plot, where the valence band edge is at -0.5 eV). These ballistic mean free path have that same exact meaning: distance traveled before a scattering event with phonons.

3) It is seen that the slowly decreasing mean free path vs. energy up to -1.6 eV in ref. 23 reproduces well their trend in Figure 2c. The kink the authors see at -1.6 eV is also seen in the mean free paths in Fig 3 of ref. 23, and it is a consequence of the fact that the ballistic mean free path depend on both the band velocity and the relaxation time, both of which depend on energy. I disagree that there is a more mysterious meaning to the data in Fig. 2c as they suggest in the manuscript.

4) This evidence suggests that toluene molecules are "manipulated" (here, desorbed?) in response to phonon emission (ie, creation of lattice vibrations), which occurs at distances from the injection point greater than the ballistic mean free paths. This should also answer the question they pose on page 5: "why the mysterious inflation region?" Carriers travel without phonon emission for 10-15 nm (see ref. 23). As they emit phonons, the lattice vibrations mediate toluene desorption. It's not electrons per se that mediate molecule desorption, since that would need a charge transfer and conformation change in response to it. Rather, it's "heat" and thus phonons that desorbs the molecules. This explains the physics behind the reduced desorption in the inflation region.

I encourage the authors to revise their explanation based on the clear agreement with the mean free path data for hole-phonon scattering in ref. 23, Fig. 3a, and the phonon emission following the scattering events inducing toluene desorption.

I have a few additional points:

1) The title says "2d charge dynamics", however the holes are injected from the STM tip with momentum with a non-zero normal component to the surface. Their experiments probes the dynamics in the directions normal to [111], but that shouldn't justify a "2D dynamics" in the title.

2) On page 5, line 8, the sentence on the reduced interaction of the carriers with the molecules should be changed. It's the phonons emitted by the hole that interact with the molecules, not the holes interacting directly with the molecules, if they are looking for an explanation in terms of coherent / ballistic mean free paths. So the interaction is there, but it's indirect

3) The word "inflation" should be removed. It's makes sense in economics, not in mesoscopic physics. There is no inflation involved here. A fast superdiffusion (ie "inflation") exists in strong laser pulses, but it shows on timescales much longer than those probed here

4) The abstract does not capture the main findings in the paper. It should stress the presence of a

region in which the carriers behave ballistically, thus suppressing toluene desorption. This is an indirect way of measuring ballistic mean free paths vs. energy for carriers injected by an STM, a highly original way to look at this process.

I strongly recommend publication in Nature Communications once these points have been addressed.

Also,

Reviewer #2 (Remarks to the Author)

In this work, K. R. Rusimova et.al report on the observation of the spatial evolution of injected hole carriers on a Si(111)7x7 surface covered with toluene molecules by using nonlocal STM. Based on their measurements and supported by an expansion-diffusion model they estimate the relevant spatial and time scale of the carriers coherent dynamics. The investigation seems to be carry out with care and are characterized by clear elements of novelty.

This is, in my opinion, an excellent work that substantially contributes to improve the scientific state-of-the-art knowledge about carriers coherent dynamics. Thus I think it deserves publication once authors clarified the following points:

1) In figure 1 the authors present a series of STM images and state: "Before hole injection there is a fairly uniform distribution of toluene molecules". However, it is not quite clear to me how this "uniform distribution" can be clearly deduced from the the frames A-C of the figure. Perhaps, rephrasing this statement avoiding the term "uniform" or giving a more clear explanation would be helpful for readers.

2) Given that the radial distance $r = 15\text{nm}$ is used in the whole manuscript to separate diffusion and inflation regimes, it would be helpful if the STM images could cover, at least, the 30 nm of diameter.

3) The discussion following Figure 3 is a too superficial especially regarding the theoretical estimation of the 10 fs inflation time. As I understood, this value was estimated by letting the wave-packet to reach a radius extension comparable with 15 nm (The suppression Region R). Being this the case, a paragraph clarifying this issue is be necessary.

4) Inflation times of 11 and 9 fs are estimated for the transport regimes $T^{\{UM\}}_{\{S3\}}$ and $T^{\{UM\}}_{\{S4\}}$ respectively. And even more dispersive values (14 and 4 fs) for UC molecules in Fig. S6. The physical mechanism behind this this effect is not discussed at all. A more deep discussion is necessary.

5) There are few typo errors that need to be fixed. For example:

page 2, line 3 - "scatting"

pag4 4, line 17- "fig 2C" instead should be "fig 2A"

We respect the comments of the reviewers thank them for their time and effort. Our responses are in italics below.

Reviewer #1 (Remarks to the Author):

1) The regime called "inflation" in the manuscript is nothing else than the ballistic transport regime in which carriers travel without scattering. The standard terminology is ballistic transport, and the manuscript should be changed accordingly. There is a whole field of ballistic electron microscopy and computation of ballistic mean free paths.

REPLY: We concur that our inflation regime corresponds to a ballistic transport regime, and have made this clearer in the manuscript. We have amended the sentence in the introductory paragraph to read: "Here we employ the nonlocal manipulation technique, at room temperature, to probe the ultrafast ballistic dynamics of the injected charge carrier, that is, while it coherently evolves from its original quantum state."

And on page 5, second discussion paragraph now reads: "after injection the probability distribution undergoes rapid ballistic expansion, or 'inflation', followed by relatively slow 2D diffusion."

We believe that the term 'inflation' accurately captures the essence of our findings where the probability distribution associated with the charge injected from the STM tip expands isotopically across the surface, rather than the more conventional picture of ballistic transport in a particular direction between two locations. So although we now make the connection to the field of ballistic transport, we feel the term inflation is appropriate to the content of this particular manuscript.

2) I completely agree that the 10-15 nm distances found in Figure 2c for the suppression region are the distance over which carriers travel without scattering. One can compare these distances with the computed energy-dependent mean free paths in silicon, in Figure 3 of reference 23 (Bernardi et al; note the shifted scale for the hole energy in the x-axis in that plot, where the valence band edge is at -0.5 eV). These ballistic mean free path have that same exact meaning: distance traveled before a scattering event with phonons.

REPLY: We now mention this agreement in the penultimate paragraph of the discussion. "We also note, again for bulk states, that the mean free paths computed for holes in Si have an approximate mean free path of 10s of nm, mirroring our surface sensitive measurements."

3) It is seen that the slowly decreasing mean free path vs. energy up to -1.6 eV in ref. 23 reproduces well their trend in Figure 2c. The kink the authors see at -1.6 eV is also seen in the mean free paths in Fig 3 of ref. 23, and it is a consequence of the fact that the ballistic mean free path depend on both the band velocity and the relaxation time, both of which depend on energy. I disagree that there is a more mysterious meaning to the data in Fig. 2c as they suggest in the manuscript.

REPLY: We have removed the offending question on page 5 "The question is: what is the physical process that generates the inflation region". Perhaps it did make our explanation seem too mysterious.

As to the reviewer's comparison of our Figure 2c and Bernardi's figure 3 we are somewhat confused as in our reading of these the trends appear to be opposite. In Bernardi's work the mean free path decreases as the hole energy drops from the Fermi level. We observe that the mean free path increases as the hole energy drops from the Fermi level. We attribute the difference to, as Bernardi states, the sensitivity of the mean-free path to the fine detail of the particular electronic state in question. Bernardi's results are for bulk states whereas we probe surface states of the reconstructed Si(111)7x7 surface. So while we now make more connection to the Bernardi work in our manuscript, we are hesitant to draw too many parallels. We have however amended the final discussion to make greater contact with Bernardi's work.

4) This evidence suggests that toluene molecules are "manipulated" (here, desorbed?) in response to phonon emission (ie, creation of lattice vibrations), which occurs at distances from the injection point greater than the ballistic mean free paths. This should also answer the question they pose on page 5: "why the mysterious inflation region?" Carriers travel without phonon emission for 10-15 nm (see ref. 23). As they emit phonons, the lattice vibrations mediate toluene desorption. It's not electrons per se that mediate molecule desorption, since that would need a charge transfer and conformation change in response to it. Rather, it's "heat" and thus phonons that desorb the molecules. This explains the physics behind the reduced desorption in the inflation region.

I encourage the authors to revise their explanation based on the clear agreement with the mean free path data for hole-phonon scattering in ref. 23, Fig. 3a, and the phonon emission following the scattering events inducing toluene desorption.

REPLY: The point the reviewer makes is interesting. It may be that the molecule is only reacting to the emission of phonons as the charge carrier decays. But it could also be a non-adiabatic process that leads to molecular manipulation. We do not feel we have sufficient data in the manuscript to unambiguously support one mechanism over the other. Moreover, the manipulation process itself is not the main focus of this work. However, we have amended the sentence on page 4 par 2, to read: "(The manipulation of the molecule, i.e. induced movement away from the original binding site, could in principle be due either to a phonon emission during this decay or a non-adiabatic decay channel \cite{Palmer1992}.)" and in accordance with point 2 below added the following clause to the sentence on page 5 line 8 " , either directly through a short lived ionic state or indirectly as the carrier decays via phonon emission,"

I have a few additional points:

1) The title says "2d charge dynamics", however the holes are injected from the STM tip with momentum with a non-zero normal component to the surface. Their experiments probes the dynamics in the directions normal to [111], but that shouldn't justify a "2D dynamics" in the title.

REPLY: The most prominent features in the STS are the surface electronic states. However, we agree that these are not in a strict theoretical sense 2D as there is an obvious coupling with the bulk (our decay channel). Therefore we have amended our title to read "... coherent surface dynamics ...".

2) On page 5, line 8, the sentence on the reduced interaction of the carriers with the molecules

should be changed. It's the phonons emitted by the hole that interact with the molecules, not the holes interacting directly with the molecules, if they are looking for an explanation in terms of coherent / ballistic mean free paths. So the interaction is there, but it's indirect

REPLY: See reply to point 4 above. Two sentences have been amended to explicitly mention that indirect phonon mediated manipulation is a possible mechanism.

3) The word "inflation" should be removed. It's makes sense in economics, not in mesoscopic physics. There is no inflation involved here. A fast superdiffusion (ie "inflation") exists in strong laser pulses, but it shows on timescales much longer than those probed here

REPLY: See earlier reply to point 1 above.

4) The abstract does not capture the main findings in the paper. It should stress the presence of a region in which the carriers behave ballistically, thus suppressing toluene desorption. This is an indirect way of measuring ballistic mean free paths vs. energy for carriers injected by an STM, a highly original way to look at this process.

REPLY: We have re-written the abstract accordingly (please excuse the LaTeX commands).

“The tip of a scanning tunnelling microscope (STM) is a source of electrons and holes and can precisely inject charge carriers into atomically well-defined sites. As the injected charge spreads out it can induce adsorbed molecules to react or desorb. By comparing large-scale ‘before and after’ STM images of a surface partly covered with molecules, the spatial extent of the nonlocal manipulation is revealed, thus allowing real-space measurement of nanometre charge transport. Here we measure the nonlocal manipulation of toluene molecules on the Si(111)-7x7 surface at room temperature. We find that both the range and probability of nonlocal manipulation per injected hole charge depend on the injection voltage. A distinct region exists within 5 to 15 nm of the injection site (dependent on voltage) where the manipulation appears to be suppressed. We propose that this region marks the extent of the initial coherent (i.e., ballistic) time-dependent evolution of the injected charge carrier. Using atomically resolved scanning tunnelling spectroscopy, we develop a model to describe this time-dependent expansion of the initially localized hole wavepacket within a particular surface state and deduce a quantum coherence (ballistic) lifetime of ~ 10 fs.”

I strongly recommend publication in Nature Communications once these points have been addressed.

Reviewer #2 (Remarks to the Author):

In this work, K. R. Rusimova et.al report on the observation of the spatial evolution of injected hole carriers on a Si(111)7x7 surface covered with toluene molecules by using nonlocal STM. Based on their measurements and supported by an expansion-diffusion model they estimate the relevant spatial and time scale of the carriers coherent dynamics. The investigation seems to be carry out with care and are characterized by clear elements of novelty.

This is, in my opinion, an excellent work that substantially contributes to improve the scientific state-of-the-art knowledge about carriers coherent dynamics. Thus I think it deserves publication once authors clarified the following points:

1) In figure 1 the authors present a series of STM images and state: "Before hole injection there is a fairly uniform distribution of toluene molecules". However, it is not quite clear to me how this "uniform distribution" can be clearly deduced from the the frames A-C of the figure. Perhaps, rephrasing this statement avoiding the term "uniform" or giving a more clear explanation would be helpful for readers.

REPLY: We have added full-scale (50 x 50 nm) images to the supplementary information as figure S1 showing the full extent of figures 1A and D. We have also reworded the text with "random distribution" rather than uniform.

2) Given that the radial distance $r = 15\text{nm}$ is used in the whole manuscript to separate diffusion and inflation regimes, it would be helpful if the STM images could cover, at least, the 30 nm of diameter.

REPLY: We are pleased to adopt this suggestion and have amended figure 1 so that it presents images of 30 x 30 nm.

3) The discussion following Figure 3 is a too superficial especially regarding the theoretical estimation of the 10 fs inflation time. As I understood, this value was estimated by letting the wave-packet to reach a radius extension comparable with 15 nm (The suppression Region R). Being this the case, a paragraph clarifying this issue is be necessary.

REPLY: See reply to point 4 below.

4) Inflation times of 11 and 9 fs are estimated for the transport regimes $T^{\text{UM}}_{\text{S3}}$ and $T^{\text{UM}}_{\text{S4}}$ respectively. And even more dispersive values (14 and 4 fs) for UC molecules in Fig. S6. The physical mechanism behind this this effect is not discussed at all. A more deep discussion is necessary.

REPLY: We have re-worked our final few paragraphs of discussion to tackle point 3 and 4 here.

"The inflation time represents the coherent or ballistic transport regime. We find no direct comparison in the literature, but the relaxation time of electrons within the dangling bond U_{-1} state has been measured to be ~ 40 fs [Mauerer2006]. This is longer than our ~ 10 fs time-scale and, may be due to the U_{-1} state lying within the bulk band-gap unlike our higher-lying states.

However, recent calculations for holes in bulk silicon giving relaxation times of ~ 10 fs \cite{Bernardi2014} have demonstrated the sensitivity to both the identity of the band that the electronic excitation resides in, and to the energy of the electronic excitation within the band. We also note, again for bulk states, that the mean free paths computed for holes in Si have an approximate mean free path of 10s of nm, mirroring our surface-sensitive measurements.

The STM injects charges with a range of energies, from the bias voltage level to the Fermi level. Figure \ref{fig_model}B shows this relationship in terms of parallel momenta. Our experiments are therefore a weighted average over a range of energies and hence, according to ref \cite{Bernardi2014}, a range of coherent lifetimes. In our simple scheme, we fit a single inflation (coherence) lifetime to each surface state. The site-to-site variation of the lifetimes we obtain may reflect the spatial distribution of the particular surface-states within a Si(111)-7x7 unit cell giving rise to differing site-to-site coherent propagation. Phonon amplitudes will also vary within the large 7x7 unit cell \cite{Liebhaber2014}, so that e-phonon relaxation rates can be expected to be site-specific.

The coherent-inflation time itself most likely has two main components, a temperature-independent coupling to bulk states and a temperature-dependent phonon scattering. Our earlier work with temperature-dependent electron injection found no obvious increase in the inflation region at lower temperatures, suggesting that the measured inflation region is an intrinsic property of the surface state. It therefore offers the possibility (as in the case of C₆₀ \cite{Nouchi1997} and other molecular overlayers \cite{Chen2009}) of designing a system with reduced coupling to bulk states and hence an increased quantum coherent inflation range, thereby extending the range over which a possible quantum device could work.”

5) There are few typo errors that need to be fixed. For example:

page 2, line 3 - "scatting"

pag4 4, line 17- "fig 2C" instead should be "fig 2A"

REPLY: Those are fixed, as are a few other minor typos that we have identified.

Reviewers' Comments:

Reviewer #1 (Remarks to the Author)

I am happy with the changes the authors made. The manuscript can be published in its current form.

Reviewer #2 (Remarks to the Author)

The authors have made necessary corrections/modifications in the revised version of the manuscript and have addressed most of the previous referee's comment. The manuscript is in a much better shape now. The changes are satisfactory, so the manuscript can now be accepted for publication.